# Association of exogenous factors with the access to innovative pharmaceutical products in Hungary

Gergő Merész[1,2]*, Péter Gaál[1]

1 Health Services Management Training Centre, Faculty of Health and Public Administration, Semmelweis University, Budapest, Hungary, 2 Department of Health Technology Assessment, National Institute of Pharmacy and Nutrition, Budapest, Hungary

* meresz.gergo@ogyei.gov.hu

## Abstract

### Introduction

Access to innovative pharmaceuticals is thought to be associated with several exogenous factors related to the local legal or financial framework of pharmaceutical reimbursement. Our aim was to describe the association between the outcome of the reimbursement procedure on innovative pharmaceutical submissions in Hungary and several potential explanatory variables related to the legal or financial framework of reimbursement procedures, such as the submission implying a need for a legal act to conclude on a positive decision; having a risk-sharing agreement (RSA) in place at the time of submission; the aim of the submission and expenditure on individual funding requests.

### Methods

Publicly available administrative announcements of the Hungarian National Health Insurance Fund Manager were used to construct the analysis dataset including all concluded procedures between 1 January 2018 and 7 June 2021, complemented with information on the overall aim of the submission (new compound or new indication). Logistic regression models were used to estimate odds ratios while adjusting for potential confounding.

### Results

Needing a legislative change as a proxy of involving high-level decision makers to reimburse had a lower (OR = 0.05, CI95%:0.02–0.11), whereas having an RSA had a statistically significant higher chance of a positive decision (OR = 3.49, CI95%:1.56–7.82). In contrast, neither the overall purpose of the submission (OR = 1.32, CI95%:0.65–2.69), nor the average biennial expenditure on individual funding requests exceeding 200 million HUFs (OR = 1.04, CI95%:0.92–1.19) had a statistically significant association with the decision.

### Conclusions

This study quantitatively demonstrated that the need for legal acts to conclude on a positive decision decreases, whereas having an RSA for the particular product increases the

**Data Availability Statement:** Data and the analysis script is available at the following repository: https://github.com/mereszgergo/coverage.

**Funding:** The authors received no specific funding for this work.

**Competing interests:** The authors have declared that no competing interests exist.

likelihood of a positive reimbursement decision in Hungary. The role of other factors remain unclear. Our findings suggest that the legal requirements and RSAs play key roles in the reimbursement of innovative pharmaceuticals and can be viewed as potential areas of policy interventions in expanding access to these products, although the feasibility of such interventions need strong commitment from decision-makers, as well as implying increased autonomy to the entities involved in reimbursement procedures. Further research is needed to assess the impact of endogenous and exogenous factors in a coherent framework.

## Introduction

Patient access to new treatments has been in the focus of a number of high-level healthcare stakeholders over past decades, with some focusing on timeliness and equality [1], while others echoing the importance of affordability [2]. Although having a tax-funded social health insurance system with a single payer, the so-called National Health Insurance Fund Manager (NHIFM), which provides practically universal coverage, drug policy in Hungary faces the similar challenges as other European countries in terms of patient access to innovative pharmaceutical treatments [3–5].

As an initial step to identify possible determinants of patient access, the local decision-making procedure must be described. The procedure to finance an innovative pharmaceutical product is initiated by the market authorization holder submitting a reimbursement dossier for a product-indication pair. The dossier contains the summary of clinical evidence and the underlying reports, as well as an economic evaluation covering the cost-effectiveness and budget impact analyses. The dossier is critically appraised using published methods [6] that are tailored to meet the local needs by the local HTA body, an non-binding assessment report is issued. The critical appraisal procedure culminates in a dedicated meeting of the Health Technology Assessment Committee, and a recommendation for decision is formulated, and initial price negotiations (i.e. arranging risk-sharing agreements) may start. If the dossier proposes to introduce a new compound, or a new indication of an already reimbursed compound to the list of reimbursed pharmaceutical products, certain ministerial decrees will need to be changed, and therefore, the Secretary of State for Health (and the Ministry of Finance) needs to be involved. Ideally, the Secretary of State convenes semi-annually a committee responsible for the prioritisation of dossiers with a positive recommendation to reimburse within the budgetary limits. The committee consists of representatives from the payer (i.e. NHIFM), the Department of Pharmaceutical Policy and Medical Devices of the Secretary of State for Health, the Ministry of Finance, as well as the Department of Health Technology Assessment at the National Institute of Pharmacy and Nutrition (NIPN). Due to the number of actors involved and the complexity of the procedure, timely decisions are needed to reimburse innovative pharmaceutical products, as otherwise patients may have limited or delayed access to these treatments [7]. The delayed access to innovative pharmaceutical products may arise from endogenous factors directly related to the clinical or economic properties of the product submitted for reimbursement (i.e. clinical added value; cost-effectiveness) or from factors outside of these domains, and rather related to the reimbursement procedure, that is, exogenous to the actual submission. Barriers of patient access related to local reimbursement procedures have been already assessed in details by some researchers for particular disease areas using qualitative data from certain Central and Eastern European countries, including Hungary [8]. Godman and colleagues [9] reported on a narrative review prepared to identify barriers for access to new medicines, touching base on a number of factors related to the features of multiple

criteria decision analysis tools and the role of managed entry agreements in enhancing chances of reimbursement for innovative products.

However, only anecdotal evidence is available on what, how and to what extent financial and procedural circumstances influence the outcome of decision-making on the submissions of innovative pharmaceutical products in Hungary. Having the list of reimbursed health technologies set out by governmental decrees can be perceived as an administrative barrier. As a consequence, submissions requiring a legislative change due to the local legal environment are expected to be less likely to conclude in a positive decision This assumption is supported by the fact that such situations require involving the Ministry of Human Capacities (the ministry responsible for health in Hungary), and decision-making continues on a higher, political level, with the involvement of further stakeholders and long bureaucratic processes. Supporting such a submission usually requires the reallocation of funding between sub-budgets or additional resources. Another frequently, but anecdotally echoed possible explanatory factor, in this case for positive reimbursement decisions, is already having a risk-sharing agreement (RSA) in place for a particular compound at the time of submitting the dossier. It is reasoned that in these situations (i.e. extending the reimbursed indications of a compound), the payer and the market authorization holder have already closed an earlier round of price negotiations, so there is less uncertainty around the financial impact of reimbursement decisions for both parties. This suggests a mutual interest in having a positive reimbursement decision, and may accelerate patient access, as suggested by others [10]. However, the aim of the submission can also be viewed as a separate explanatory variable alongside RSAs. Submissions aiming to introduce an entirely new compound (or a new combination of compounds) can potentially be distinguished from those that propose the introduction of a new indication for an already reimbursed compound. An entirely new compound would possibly invoke greater interest, as it may address an unmet need in the population or offer a new therapeutic option for patients and clinicians. A somewhat different situation can occur if a new indication of an already reimbursement compound is requesting reimbursement. It is possible that experience with financing the treatment may impact the outcome of the decision regardless of a valid RSA. Reimbursement submissions may also have other purposes that are less relevant to the current research, such as adjusting the reimbursement technique (i.e. moving the product from the inpatient setting to the outpatient setting), introducing a new pharmaceutical form, or increasing the price of an already reimbursed compound. Finally, higher levels of expenditure on individual funding requests (IFRs) may also motivate the payer to conclude on a positive decision as handling these requests can be burdensome in both administrative and financial terms. IFRs are a system to provide reimbursement for products and indications, which have not been included in the positive list of pharmaceuticals of the public benefit package, yet, but there is some evidence of clinical benefit for (usually a small number of) patients. There is no quantitative restriction on the number IFRs submitted by clinicians or any strong barrier alike, and the documentation supporting each request that needs to be attached is quite brief as well, resulting in an acceptance rate of over 88% in 2020 [11]. Hence, they are an exceptionally useful means of enabling health equity and patient access, as submitting them are in the merit of the patient and the clinician. However, the number of such requests (around 20,000 per annum in the past years) and the limited bargaining power of the financing agent for this financing mechanism can incentivise the healthcare decision-makers to transfer products with high IFR volumes into routine reimbursement as soon as possible. Moreover, pharmaceutical products that have relatively high IFR volumes consequently have a high level of baseline spending, and the formal reimbursement of decision comes with only minor net budget impact on the level of the healthcare budget. Hence it is plausible to think that having an IFR contributes to a positive reimbursement decision.

Although the availability of innovative pharmaceutical products is monitored regularly in European countries, the factors contributing to limited or delayed access have so far been mostly explored with qualitative techniques. The aim of the current study is to provide a quantitative description of the association between the outcome of reimbursement decision with the aforementioned potential contributing exogenous factors.

## Materials and methods

In order to explore the association between the outcome of the reimbursement procedure, the anecdotal contributing factors were operationalised as independent variables in a multivariate logistic regression framework. Univariate- and multivariate models were used to estimate odds ratios (ORs), quantifying the association between each factor and the outcome of the decision procedure. Basic information is publicly available on each reimbursement submission [12], maintained by the NHIFM. Apart from the name of the product, date of the submission and further administrative details, this data source contains the outcome of the procedure and information on whether a positive decision on the submission would imply the amendment of the current legislation.

The need for the amendment of the current legislation occurs, if a new compound is submitted for reimbursement, aiming at an indication which is not yet recognized in legal acts. This essentially means that the Ministry of Human Resources is directly involved as a new entity in the decision-making procedure. This variable is expected to decrease the chance of a positive decision on reimbursement.

Having an RSA in place on the submitted product at the time of submission means that the NHIFM and the market authorization holder (MAH) already engaged in price negotiations before the actual submission. This is mostly relevant for submissions with purposes other than introducing an entirely new compound to the healthcare system. This variable is expected to increase the risk of a positive decision on reimbursement. Although the contents of RSAs are confidential, there are public semi-annual reports on products that are reimbursed with RSAs in place [13]. This information can be used to identify submissions which were involved in an RSA at the time of their submission.

The purpose of the submission may also impact the outcome of the decision-making procedure and therefore should be assessed. The archives of the Department of Health Technology Assessment at the National Institute of Pharmacy and Nutrition were used to review the submitted documentation to identify the overall purpose of the submission, namely, whether it aimed for the reimbursement of an entirely new compound, or had other purposes, such as extending the current reimbursement of a compound to a new indication.

Finally, expenditure on IFRs may also have an impact on the reimbursement decisions, as the financing agent may not be fully capable of engaging in price negotiations, and the administrative burden of handling these requests is also demanding. Therefore, a high level of expenditure on IFRs would presumably develop an interest in the financing agent to include the product on the reimbursement list, i.e. increasing the chance of a positive decision while reducing the administrative burden. We used publicly available financial reports on expenditures reimbursed through individual funding requests of pharmaceutical products to further expand our dataset [14].

### Data analysis

As the current legislative framework of pharmaceutical reimbursement submissions was issued in late 2017 [15], reimbursement submissions between 1 January 2018 and 7 June 2021 were considered for this analysis. First, dossiers not submitted for the full procedure and procedures

which did not conclude during the observation period were both omitted from the analysis. Second, procedures related to submissions aiming to increase the price of an already reimbursed product were also excluded, as these are not relevant to providing information on access to innovative pharmaceutical products.

Apart from descriptive analyses, the association between the outcome of the decision-making procedure as the dependent variable and the need of a legal act, having an RSA in place at the time of submission, overall purpose of the submission or expenditure on IFRs as independent variables was studied in univariate and multivariate logistic regression framework. A quasi case-control study design was adapted where the outcome of the decision-making procedure was coded as a binary variable with the value „1" marking a positive decision, and „0" if any other outcome was reached (negative decisions or no decision reached within mandatory deadlines of the procedure). The need of a legal act for decision-making (yes = 1, no = 0), the presence of an RSA at the time of submission (yes = 1, no = 0), overall purpose of the dossier (new compound or a new combination containing a new compound = 1, any other purpose, but not a price increase = 0) were also included as dichotomous variables. Expenditures reimbursed on the basis of IFRs were collected for the actual and preceding year of submission for each procedure. However, to help interpretation, this variable was recoded by using units of 200 million HUFs for the biennial expenditures on IFRs to tackle the potential seasonality and fluctuation of expenditures on IFRs. Data analysis was conducted in R version 4.1.2 [16]. The threshold for statistical significance was 0.05. Missing data points were omitted from the analyses.

## Results

The total number of reimbursement submissions between 1 January 2018 and 7 June 2021 was 1,390. Among these submissions, 486 was submitted in the full procedure, of which 162 did not conclude at the time of analysis. Of the remaining 324 submissions, 92 was aiming for a price increase, or adjusting the reimbursement technique, and were considered irrelevant to the current research.

The primary analysis dataset consisted of information on 232 submissions, enriched with data on legal act needed for the decision, RSAs in place at the time of submission, compound-level biennial expenditure on IFRs (expressed HUFs) and overall aim of the submission.

Table 1 summarises the descriptive data on submissions according to the outcome of the decision-making and independent variables. Chi-square and independent samples T-tests were performed to study the association between the outcome of the decision-making procedure and the independent variables.

More, than two thirds of reimbursement submissions with a positive decision did not need a legal act, whereas 86.82% of reimbursement dossiers with non-positive outcomes would have needed a modification of the legal acts, yielding a statistically significant difference. The proportion of dossiers where the compound itself was covered by an RSA at the time of submission were also significantly more frequent in case of procedures with positive outcomes. Among reimbursement procedures with a non-positive outcome, dossiers proposing to introduce a new compound (or a combination with a new compound) to the healthcare system were also more frequent, although statistically not significant. The average biennial expenditure on the compound, based on IFRs tended to be higher in the case of procedures which finally concluded in a positive decision, yet the difference was not statistically significant.

The results of the univariate and multivariate logistic regression models (see Table 2) showed a consistent significant negative association between needing a legal act for the positive decision and the log odds of the procedure arriving at a positive decision (adjusted OR = 0.05,

**Table 1. Description of the submissions according to independent variables.**

| | Positive | Any other outcome | All procedures | p-value |
|---|---|---|---|---|
| A legal act is needed for the positive decision | | | | |
| Yes | 30 (29.13%) | 112 (86.82%) | 142 (61.21%) | p<0.001 |
| No | 73 (70.87%) | 17 (13.18%) | 90 (38.79%) | |
| RSA was in effect at the time of submission | | | | |
| Yes | 32 (31.07%) | 18 (13.96%) | 50 (21.55%) | p = 0.002 |
| No | 71 (68.93%) | 111(86.04%) | 182 (78.44%) | |
| Overall aim of the submission | | | | |
| Introducing a new compound or a combination with a new compound | 13 (12.62%) | 53 (41.09%) | 66 (28.21%) | p<0.001 |
| Introducing a new indication for an already reimbursed compound | 48 (46.60%) | 70 (54.26%) | 118 (50.43%) | |
| Other, but not a price increase | 42 (40.78%) | 6 (4.65%) | 50 (21.37%) | |
| Average biennial expenditure on compound, based on IFRs (HUF, mean (SD)) | 194,934,323 (531,420,827) | 164,388,885 (463,476,416) | 181,050,033 (499,868,965) | p = 0.5617 |

CI95%: 0.02–0.11, p<0.001). Similarly, having an RSA in place at the time of submission was consistently positively associated in both univariate and multivariate models with a statistically significant higher chance of a positive decision (adjusted OR = 3.49, CI95%: 1.56–7.82, p = 0.003). However, the overall purpose of the submission did not have statistically significant association with the outcome of the decision procedure (adjusted OR = 1.32, CI95%: 0.65–2.69, p = 0.45) and the direction of the association was not consistent between univariate and multivariate models. Neither did the average biennial expenditure exceeding 200 million HUFs show statistical significance, although it was positively associated with the decision outcome being supportive (adjusted OR = 1.04, CI95%: 0.92–1.19, p = 0.54) in both univariate and multivariate models.

As the current analysis covers a broader time period, the effect of the year of submission was included in a supplementary logistic regression analysis to explore the effect of adjusting to the date of submission. Although having the reimbursement dossier submitted in the year 2019 was associated with a statistically significant higher likelihood of a positive decision (adjusted OR = 3.24, CI95%: 1.36–7.72, p = 0.008) and therefore can be interpreted as an effect modifier, results on other independent variables were consistent both in terms of the direction of association and statistical significance with the former multivariate analysis. The goodness-of-fit characteristic was somewhat improved for the multivariate model including the year of submission (AIC: 229.11).

## Discussion

The a priori expectations on the relationship between legal and financial factors and the outcome of the reimbursement procedure have been confirmed by the data for the impact of needing a legal act for the positive decision and the impact of having an RSA in place for the

**Table 2. Results of the logistic regression analyses with the outcome of the decision being positive as the dependent variable.**

| | The decision procedure is positive | | | | | |
|---|---|---|---|---|---|---|
| | OR (CI95%) Univariate Model | | | OR (CI95%) Multivariate Model | | |
| A legal act is needed for the positive decision | 0.06 (0.03–0.12) | p<0.001 | AIC: 237.67 | 0.05 (0.02–0.11) | p<0.001 | AIC: 232.15 |
| RSA was in place at the time of submission | 2.78 (1.45–5.32) | p = 0.002 | AIC: 312.78 | 3.49 (1.56–7.82) | p = 0.003 | |
| Introducing a new compound or a combination with a new compound | 0.74 (0.44–1.24) | p = 0.25 | AIC: 321.35 | 1.32 (0.65–2.69) | p = 0.45 | |
| Average biennial expenditure on the basis of IFRs exceeds 200 million HUFs | 1.02 (0.92–1.13) | p = 0.71 | AIC: 322.56 | 1.04 (0.92–1.19) | p = 0.54 | |

particular compound at the time of submitting the reimbursement dossier. However, in case of the overall purpose of the submission and the expenditure on IFRs, results were not conclusive. A possible explanation to the former, to some extent, is that decision on a reimbursement submission that proposes the introduction a new compound may also be affected by other, endogenous factors, such as unmet medical need. It is also unclear whether the number of IFRs have a bigger impact on the outcome of the decision than the actual expenditure through the financial mechanism, but there are no publicly available data for this investigation.

In certain settings, such as for France or Germany, assessing the determinants of coverage decisions is redundant to some extent, as the outcome of the HTA procedure is directly linked to the pricing and reimbursement status of pharmaceutical products. However, in other settings, Maynou and Carnes [17] evaluated a set of factors that can potentially contribute to coverage decision from an econometric perspective and concluded for cancer drugs that their probability of reimbursement in 6 countries was higher when the product was considered cost-effective by NICE or SMC and when there was a managed entry agreement of financial type.

A strength of the current research is that attempts to explore the association between the outcome of the reimbursement procedure and certain independent variables which were generated from the legal and financial environment of each submission in a quantitative way. So far, only limited, descriptive research has been available on the reimbursement decisions in Hungary, and these did not assess the factors influencing the outcomes of decisions [18]. Further examples from other settings include using qualitative data sources to retrospectively evaluate the impact of RSAs on reimbursement decisions [19]. This study assesses multiple factors with their relevance to stakeholders involved in the decision-making procedure. Moreover, the current research applied formal hypothesis testing in a multivariate framework.

A potential methodological limitation of the current study is that we did not consider the time necessary for arriving at a decision for each procedure like other studies did [20]. Although time is certainly an important factor, we argue that the time required for decision-making may be biased in this particular setting. Positive decisions may accumulate over time, because some reimbursement procedures (such as the ones needing legislative changes or re-allocation of funds between government functions) apply a sequential, interactive approach and involve stakeholders other governmental entities (Ministry of Finance, Ministry of Interior) than the payer and the MAH. Therefore, a clear political mandate to reimburse may also be needed for such decisions that might affect the semi-annual convention of the committee responsible for the prioritisation of dossiers; a lack of mandate essentially means that committee meetings are delayed, creating a backlog in formal decision-making. This study is also limited by the operationalisation of the explanatory variables, which is determined to a great extent by the level of transparency and reporting standards of information, especially on RSAs. At the moment, the existence of RSAs attached to pharmaceutical products are public, with a categorical description of the type of manufacturer payment (simple discount, price-volume, outcome-based schemes). A further limitation to this study is that it does not assess important potential explanatory variables that are endogenous factors of submissions, for example, the conclusion on their clinical added benefit or their cost-effectiveness, although these factors can be seen as confirmed contributors to recommendations on cancer drugs for a number of agencies, as concluded by a recent systematic literature review [21]. Unfortunately, a conclusion on the clinical added benefit was not implemented in the health technology assessment procedure in Hungary at the time of the implementation of the current research. A potential direction for future research is to evaluate the association between the outcome of reimbursement processes and exogenous as well as endogenous factors in a common, coherent framework.

While the place of our research has been confined to Hungary, we consider the findings of our study relevant to many other countries, since the exogenous factors investigated in the

frame of this research are by no means exclusive to the Hungarian system of the reimbursement of pharmaceuticals [20, 22, 23]. Regardless of the local setting, it is important to note that policy tools for ensuring access are not mutually exclusive: from a broader, pharmaceutical policy perspective, this research implies that the legal framework of reimbursement decisions, as well as RSAs both play key roles in managing access to innovative products. Re-defining the situations with a clear need for legislative changes to conclude on positive reimbursement decisions is a potential way of enabling access. Otherwise, given that high-level legislative control is maintained over the reimbursement of novel compounds, the timeliness of these decisions should be ensured to enhance the predictability of decisions. Also, providing public payers with the appropriate authorisation and sufficient resources to setting up RSAs during price negotiations may broaden access to innovative treatments, while still adhering to budgetary limits. The need for timely decisions and empowering payers are both in line with others' call for defining simple, transparent and robust decision-making frameworks to strengthen pricing and reimbursement systems that increase access to innovative pharmaceutical products while also ensuring affordability [24].

One of the lessons learned from the current research is that when assessing the delay in access to innovative pharmaceutical products, a distinction should be made between submissions that require a sequence of decisions (such as implementing new legislative acts) for granting access, and submissions that only require lower level decision making in order to reach a conclusion on reimbursement. Another distinction could be made based on whether the product of interest has been included in an RSA between the financing agent and the MAH. This should be for particular interest in the near future as according to a new legislation, RSAs may also be used to manage expenses based on IFR applications of reimbursed compounds as of the 28 June 2021 [25]. Therefore, a future research may also consider the cases of compounds with a valid market authorization that were not formally submitted for reimbursement, but patients may have limited access to them via IFRs and RSAs. However, when evaluating the role of RSAs in broadening access, their administrative burden, transparency and effectiveness as cost-containment measures should also be taken into account.

## Conclusion

The current research provides a quantitative analysis of the association between certain explanatory variables depicting the local health financing environment of pharmaceutical reimbursement submissions and the outcome of the corresponding decision-making procedure in Hungary. In this study, it was shown that the need of a legal act statistically significantly independently decreases, whereas having an RSA in place at the time of submission significantly increases the likelihood of a positive decision. The role of other factors, such as the purpose of the submission or the amount of expenditure through named patient programmes, do not have a statistically significant impact on the decision outcome, although the latter seems to have a positive association with a positive decision.

Our findings suggest that the local legal and financial environment of pharmaceutical policy has a significant impact on patient access to innovative pharmaceuticals. This paper may contribute to highlighting the potential targets of pharmaceutical policy interventions to expand patient access to innovative health technologies. To complement the picture, further research is needed on the role of endogenous factors and their association with the factors studied here.

## Author Contributions

**Conceptualization:** Gergő Merész, Péter Gaál.

**Data curation:** Gergő Merész.

**Formal analysis:** Gergő Merész.

**Investigation:** Gergő Merész.

**Methodology:** Gergő Merész, Péter Gaál.

**Supervision:** Péter Gaál.

**Writing – original draft:** Gergő Merész, Péter Gaál.

**Writing – review & editing:** Gergő Merész, Péter Gaál.

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
