## [Decision Letter · Decision Letter 0]

26 Jul 2022

PONE-D-21-40982Association of Exogenous Factors with the Access to Innovative Pharmaceutical Products in HungaryPLOS ONE

Dear Dr. Merész,

Thank you for submitting your manuscript to PLOS ONE. After careful consideration, we feel that it has merit but does not fully meet PLOS ONE’s publication criteria as it currently stands. Therefore, we invite you to submit a revised version of the manuscript that addresses the points raised during the review process.

Please note that we have only been able to secure a single reviewer to assess your manuscript. We are issuing a decision on your manuscript at this point to prevent further delays in the evaluation of your manuscript. Please be aware that the editor who handles your revised manuscript might find it necessary to invite additional reviewers to assess this work once the revised manuscript is submitted. However, we will aim to proceed on the basis of this single review if possible.  The reviewer requests a more in-depth discussion of the studies premise, results and the associated literature. Could you please revise the manuscript?

We look forward to receiving your revised manuscript.

Kind regards,

Thomas Tischer

Staff Editor

PLOS ONE

Journal Requirements:

2. Please note that in order to use the direct billing option the corresponding author must be affiliated with the chosen institute. Please either amend your manuscript to change the affiliation or corresponding author, or email us at plosone@plos.org with a request to remove this option.

Reviewers' comments:

Reviewer's Responses to Questions

**Comments to the Author**

1. Is the manuscript technically sound, and do the data support the conclusions?

Reviewer #1: Partly

2. Has the statistical analysis been performed appropriately and rigorously? 

Reviewer #1: I Don't Know

3. Have the authors made all data underlying the findings in their manuscript fully available?

Reviewer #1: No

4. Is the manuscript presented in an intelligible fashion and written in standard English?

Reviewer #1: Yes

5. Review Comments to the Author

Reviewer #1: This is an interesting piece of work that adds to the literature on determinants of access to medicines in a particular setting (Hungary) - thank you for working in this area and preparing your manuscript for publication. My main reservations rest with a) the lacking description of the decision-making process in Hungary that would elucidate what could potentially be changed and how; b) the partial contextualisation within the broader literature on determinants of coverage decisions; and c) the need for a more nuanced discussion of the lessons learned (is the article really advocating for managed entry agreements for all new medicines? at least the issue of administrative burden, transparency and cost containment should be discussed). Please see the following comments for more detail on suggested changes.

Abstract

The independent variables need to be more clearly named. The "need for legilative change to reimburse" is not clear without the context of the Hungarian reality that only comes in the main text. Part of the conclusions (the recommendaitons) does not appear equivalently in the main text. The recommendations should consider feasibility (see general comments above).

Introduction

When references to the literature are made (e.g. lines 46-61), it is not clear if you refer to all literature examining the relation between endogenous/exogenous factors and reimbursement decisions or only literature on Hungary. This is important to do very clearly, because there is quite a bit of research on what influences reimbursement decisions in different health systems, which is not captured in the introduction but is important for context. Please add a more comprehensive summary of available evidence (in general and in HUngary in particular) to also help set up a discussion of what this study adds later in the manuscript.

Lines 67-71: how is this different from the distinction you make in lines 62-67? The existence of RSA?

Lines 76-77: please justify the assessment of the usefulness of IFR (you will need to explain better what they are and how they fit in the HUngarian reimbursement system)

Generally for the introduction: Please provide an overall description of how reimbursement of new mediicnes works in Hungary (one paragraph, not very long) that you can use as a basis for details brought up and used in the methodology (such as need for legislative change or IFRs). This is necessary for understanding your paper and how it fits/what it adds better. You could also add a figure if helpful.

Methods

Generally, information on Managed Entry Agreements is not publicly available, but you seem to have at least some in Hungary. Please makes this clear early on tin the methods.

Lines 107-116: please check redundancy with the introduction. If you end up providing a foundational description as suggested above, you can be shorter in the methods when it comes to the definition of your variables.

Lines 123-124: this is not the place for a funding statement of your work, should be moved to the peripherals of the manuscript.

Lines 144-145 "however, this variable was recorded...200 million HUFs". This needs further explanation.

Line 152: "changing the reimbursement technique" - what does this mean? might be solved with a better description of processes in the intro as suggested above.

Results

Lines 188-192: the logic of what is in the table and what is reported only is not always clear. Is the year of submission an independent variable?

Discussion

Lines 198-200: you should not bring in new potential independent variables to discard at this point, this should be done earlier. Also perhaps spend a few words in the introduction to discuss how you categorise endogenous and exogenous parameters.

Line 203: I would argue that this is rather an attempt to explore the association in a quantitative way rather than quanitfying the association

Lines 205-210: as int he introduction, unclear if the contrast here is to literature on these issues in general or only literature on Hungary.

213-214: Please clarify this and link to a fundamental description of decision-making processes in Hungary.

Lines 232-237: this needs further explanation (what is the new law on/what does it say-how does it change the situation)

The limitations of the study need to be discussed in further detail, including the chosen methodology and the adequacy of the independent variables. Why this article is of interest beyond Hungary should be substantiated further.

Conclusions

The recommendations in the abstract do not appear in the main text (see also comment above). The potential targets (meaning legislative change and more MEAs?) should be justified in more detail.

6. PLOS authors have the option to publish the peer review history of their article (what does this mean?). If published, this will include your full peer review and any attached files.

Reviewer #1: No

---

## [Author Response · Author response to Decision Letter 0]

30 Nov 2022

Dear Reviewer,

We would like to thank you for your valuable insights and we hope that our responses address your reservations. Let us know if there is anything else to adjust on the manuscript.

Best wishes,

The Authors

---

## [Decision Letter · Decision Letter 1]

20 Jan 2023

Association of Exogenous Factors with the Access to Innovative Pharmaceutical Products in Hungary

PONE-D-21-40982R1

Dear Dr. Merész,

We’re pleased to inform you that your manuscript has been judged scientifically suitable for publication and will be formally accepted for publication once it meets all outstanding technical requirements.

Kind regards,

Vasileios Kallinterakis

Academic Editor

PLOS ONE

Additional Editor Comments (optional):

Reviewers' comments:

Reviewer's Responses to Questions

**Comments to the Author**

1. If the authors have adequately addressed your comments raised in a previous round of review and you feel that this manuscript is now acceptable for publication, you may indicate that here to bypass the “Comments to the Author” section, enter your conflict of interest statement in the “Confidential to Editor” section, and submit your "Accept" recommendation.

Reviewer #2: (No Response)

Reviewer #3: All comments have been addressed

2. Is the manuscript technically sound, and do the data support the conclusions?

Reviewer #2: Yes

Reviewer #3: Yes

3. Has the statistical analysis been performed appropriately and rigorously? 

Reviewer #2: Yes

Reviewer #3: Yes

4. Have the authors made all data underlying the findings in their manuscript fully available?

Reviewer #2: Yes

Reviewer #3: Yes

5. Is the manuscript presented in an intelligible fashion and written in standard English?

Reviewer #2: No

Reviewer #3: Yes

6. Review Comments to the Author

Reviewer #2: This research study is indeed very interesting and is conducted in a rigorous scientific way. However, in order to be published, the manuscript requires heavy editing to make the story flow better and ensure good use of the English language.

Reviewer #3: (No Response)

7. PLOS authors have the option to publish the peer review history of their article (what does this mean?). If published, this will include your full peer review and any attached files.

Reviewer #2: No

Reviewer #3: No

---

## [Editor Report · Acceptance letter]

25 Jan 2023

PONE-D-21-40982R1 

Association of Exogenous Factors with the Access to Innovative Pharmaceutical Products in Hungary 

Dear Dr. Merész:

I'm pleased to inform you that your manuscript has been deemed suitable for publication in PLOS ONE. Congratulations! Your manuscript is now with our production department. 

Kind regards, 

on behalf of

Dr. Vasileios Kallinterakis 

Academic Editor

PLOS ONE